# Hydroxychloroquine-mediated inhibition of SARS-CoV-2 entry is attenuated by TMPRSS2

**Tianling Ou\*, Huihui Mou, Lizhou Zhang, Amrita Ojha, Hyeryun Choe, Michael Farzan \***

Department of Immunology and Microbiology, The Scripps Research Institute, Jupiter, Florida, United States of America

\* tou@scripps.edu (TO); mfarzan@scripps.edu (MF)

**Data Availability Statement:** All relevant data are within the manuscript and the Supporting Information file.

**Funding:** This work is supported by an emergency administrative supplement to NIH R01 AI129868.

## Abstract

Hydroxychloroquine, used to treat malaria and some autoimmune disorders, potently inhibits viral infection of SARS coronavirus (SARS-CoV-1) and SARS-CoV-2 in cell-culture studies. However, human clinical trials of hydroxychloroquine failed to establish its usefulness as treatment for COVID-19. This compound is known to interfere with endosomal acidification necessary to the proteolytic activity of cathepsins. Following receptor binding and endocytosis, cathepsin L can cleave the SARS-CoV-1 and SARS-CoV-2 spike (S) proteins, thereby activating membrane fusion for cell entry. The plasma membrane-associated protease TMPRSS2 can similarly cleave these S proteins and activate viral entry at the cell surface. Here we show that the SARS-CoV-2 entry process is more dependent than that of SARS-CoV-1 on TMPRSS2 expression. This difference can be reversed when the furin-cleavage site of the SARS-CoV-2 S protein is ablated or when it is introduced into the SARS-CoV-1 S protein. We also show that hydroxychloroquine efficiently blocks viral entry mediated by cathepsin L, but not by TMPRSS2, and that a combination of hydroxychloroquine and a clinically-tested TMPRSS2 inhibitor prevents SARS-CoV-2 infection more potently than either drug alone. These studies identify functional differences between SARS-CoV-1 and -2 entry processes, and provide a mechanistic explanation for the limited *in vivo* utility of hydroxychloroquine as a treatment for COVID-19.

## Author summary

The novel pathogenic coronavirus SARS-CoV-2 causes COVID-19 and remains a threat to global public health. Chloroquine and hydroxychloroquine have been shown to prevent viral infection in cell-culture systems, but human clinical trials did not observe a significant improvement in COVID-19 patients treated with these compounds. Here we show that hydroxychloroquine interferes with only one of two somewhat redundant pathways by which the SARS-CoV-2 spike (S) protein is activated to mediate infection. The first pathway is dependent on the endosomal protease cathepsin L and sensitive to hydroxychloroquine, whereas the second pathway is dependent on TMPRSS2, which is unaffected by this compound. We further show that SARS-CoV-2 is more reliant than SARS coronavirus (SARS-CoV-1) on the TMPRSS2 pathway, and that this difference is due to a furin

The funders had no role in the study design, data collection and analysis, decision to publish, or preparation of this manuscript.

**Competing interests:** The authors have declared that no competing interests exist.

cleavage site present in the SARS-CoV-2 S protein. Finally, we show that combinations of hydroxychloroquine and a clinically tested TMPRSS2 inhibitor work together to effectively inhibit SARS-CoV-2 entry. Thus TMPRSS2 expression on physiologically relevant SARS-CoV-2 target cells may bypass the antiviral activities of hydroxychloroquine, and explain its lack of *in vivo* efficacy.

## Introduction

The pandemic coronavirus disease 2019 (COVID-19), caused by SARS coronavirus 2 (SARS-CoV-2) poses serious threat to global public health [1]. In the first months following the onset of the pandemic, several existing drugs were reconsidered for COVID-19 treatments, among them chloroquine and its derivative hydroxychloroquine sulfate (hydroxychloroquine) [2]. The use of hydroxychloroquine has become controversial as clinical trials suggest that this drug is ineffective as either a treatment or a prophylaxis against SARS-CoV-2 infection. The United States Food and Drug has since revoked its emergency use authorization for this drug [3–7]. This disappointing result contrasts with its promising cell culture studies which demonstrated a half-maximal effective concentration ($EC_{90}$) of 6.9 μM against SARS-CoV-2 replicating in Vero E6 cells. This concentration is achievable *in vivo* with a well-tolerated 500 mg daily administration and similar with the $EC_{50}$ of the relatively more successful remdesivir (1.76 μM) [8–10].

Hydroxychloroquine has been suggested to restrict multiple steps in the coronaviral lifecycle [11–13], but its inhibitory effect on viral entry as a lysosomotropic agent is best defined [14]. It elevates endosomal pH and subsequently interrupts activities of cathepsin L, one of the entry factors for coronaviruses [15,16]. Coronaviral entry requires both receptor engagement and fusion activation by proteolytic processing the S glycoproteins. The SARS-CoV-1 and -2 S proteins bind angiotensin-converting enzyme 2 (ACE2), their common receptor [17–19]. Two obligate proteolysis sites for fusion activation have been identified within the S proteins, namely at the junction of the S1 and S2 domains, and at the S2' site in an exposed loop of the S2 domain [20]. The SARS-CoV-2 S1/S2 junction is cleaved in virus producing cells by proprotein convertases that cleave a distinctive furin-recognition site at this boundary [21]. In contrast, the SARS-CoV-1 S1/S2 boundary is cleaved in the virus target cell after receptor engagement by either cell-surface TMPRSS2 or endosomal cathepsin L [15,22,23]. The S2' sites of both viruses are cleaved in the target cells, again by either TMPRSS2 or cathepsin L (**Fig 1**). These proteolysis events and ACE2-binding prime the S protein for conformational changes that mediate fusion between the viral and cellular membranes [22–25]. While hydroxychloroquine is known to suppress cathepsin proteolysis activity, its impact on TMPRSS2-mediated viral entry is unknown.

Cells that express both ACE2 and TMPRSS2 are present in multiple tissues including lung (alveolar and bronchial), buccal mucosa, nasal mucosa, ileum, colon, and myocardium epithelium [23,26]. Recent studies of single-cell RNA sequencing further locate these highly susceptible cells in respiratory tree, cornea, esophagus, ileum, colon, gallbladder and common bile duct [27]. Three major cell types are identified to have TMPRSS2 and ACE2 co-expression: lung type II pneumocytes, ileal absorptive enterocytes, and nasal goblet secretory cells [28]. Of note, while ACE2 has a generally lower expression level and a narrower distribution than TMPRSS2, most ACE2-positive cells in the respiratory tract also express TMPRSS2 [27–31].

Here we evaluated the infectivity of SARS-CoV-1 and -2 S proteins on cells in the presence and absence of TMPRSS2. We show that TMPRSS2 expression has a markedly greater impact on entry mediated by the SARS-CoV-2 S protein than by that of SARS-CoV-1. We further show that antiviral efficiency of hydroxychloroquine on SARS-CoV-2 S-protein-mediated

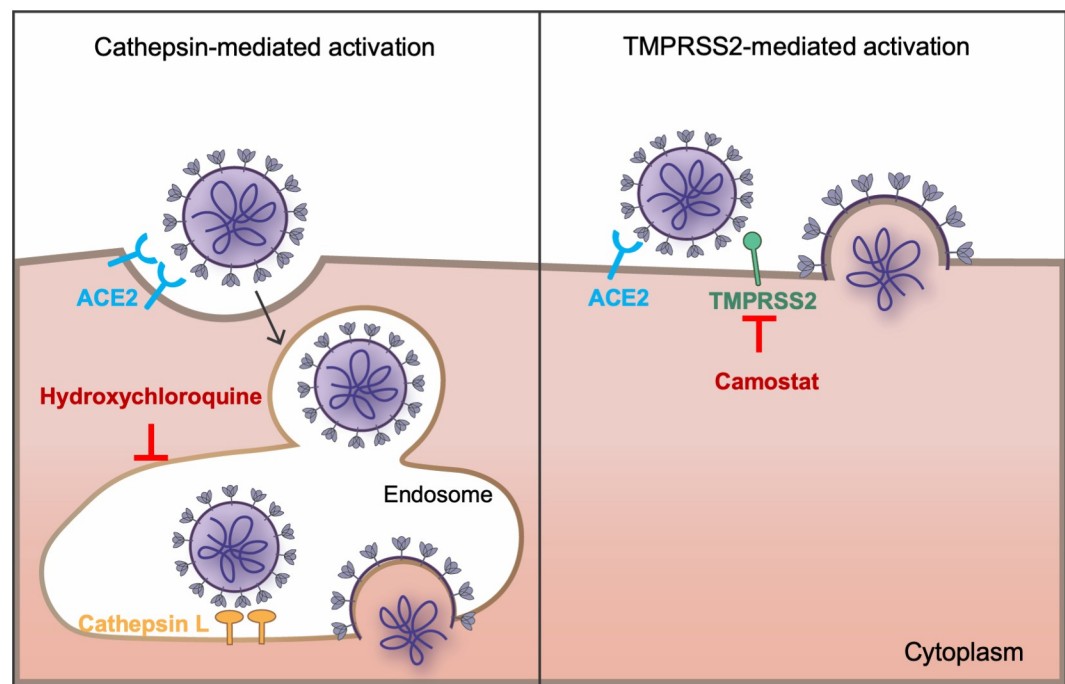

**Fig 1. SARS-CoV-1 and SARS-CoV-2 fusion can be activated by either or both of two pathways.** The coronaviruses bind the cellular receptor ACE2 and must be activated by proteolysis with either a surface-expressed protease like TMPRSS2 or by cathepsin L in the endosome. Only cathepsin L-mediated proteolysis requires endosomal acidification. Camostat mesylate inhibits TMPRSS2 activity, whereas hydroxychloroquine, like ammonium chloride, inhibits endosomal acidification.

entry is negatively impacted by the expression level of TMPRSS2. Consistent with these observations, when cells are expressing a high level of TMPRSS2, the TMPRSS2 inhibitor camostat was more potent than hydroxychloroquine at inhibiting SARS-CoV-2 infection, but the converse for SARS-CoV-1. Finally, we demonstrate that the anti-SARS-CoV-2 activity of hydroxychloroquine could be enhanced by camostat but not by compounds that inhibit cathepsin L. Thus failure of hydroxychloroquine in clinical studies may reflect the presence of TMPRSS2 in key tissues and its importance to the SARS-CoV-2 entry process.

## Results

### TMPRSS2 allows efficient cell entry mediated by the SARS-CoV-2-S protein, bypassing a cathepsin L-dependent endosomal entry pathway

TMPRSS2 can proteolytically activate membrane fusion of a variety of respiratory viruses including influenza A virus, SARS-CoV-1, MERS-CoV and SARS-CoV-2 [22–24,32,33]. Overexpression of TMPRSS2 was shown to increase the susceptibility of cells to MERS-CoV [32]. We first investigated how TMPRSS2 expression affected SARS-CoV-2 infectivity. To do so, we measured the viral entry of pseudovirions (PV) bearing SARS-CoV-1, SARS-CoV-2 S, or vesicular stomatitis virus (VSV) G proteins (SARS1-PV, SARS2-PV, VSV-PV). HEK293T-ACE2 cells (a stable cell line) transiently transfected with TMPRSS2 or control plasmids (mock) were used as target cells. As demonstrated in **Fig 2A**, when TMPRSS2 was expressed, viral entry mediated by the SARS-CoV-2-S protein was significantly increased (up to 100-fold). In contrast, enhancement of SARS1-PV entry by TMPRSS2 was less pronounced (0.5–10 fold), and there was no enhancement of VSV-PV entry from TMPRSS2. We then asked whether

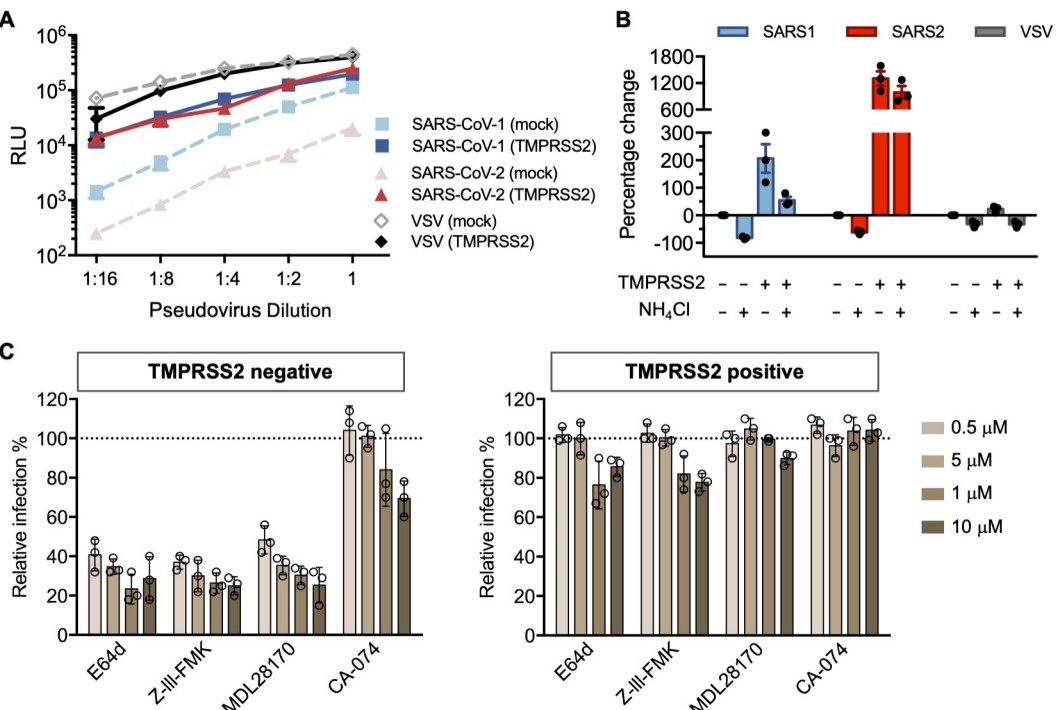

**Fig 2. SARS-CoV-2 is resistant to endosomal protease inhibitors in cells over-expressing TMPRSS2. (A)** 293T-ACE2 cells were transfected with a vector control plasmid (TMPRSS2-negative), or a TMPRSS2 expression plasmid, 24 hours prior to infection. Retroviruses pseudotyped with SARS-CoV-1 (SARS1), SARS-CoV-2 (SARS2) S, and VSV G proteins were used for infection at titers yielding equivalent infection in the presence of TMPRSS2. Pseudovirus was serially diluted by 2-fold starting from stock (RLU≈250,000 for TMPRSS2-expressing cells). Cells were inoculated with diluted pseudovirus, and viral entry was determined by the luciferase activity in cell lysates within 48 hours post infection. Shown is a representative plot from three experiments. Each point indicates the mean (±SD) of duplicate samples. **(B)** Cells were treated by 50 mM of ammonium chloride before infection. Infection of TMPRSS2-negative cells without treatment was used for normalization. **(C)** A panel of cathepsin inhibitors was tested: E64d and Z-III-FMK inhibit both cathepsin B and cathepsin L; MDL28170 specifically inhibits cathepsin L, and CA-074 inhibits cathepsin B. Cells were treated with the indicated concentrations of protease inhibitors or DMSO for 2 hours, then inoculated with retrovirus harboring SARS-CoV-2 spike proteins. Luciferase activity was measured at 48 hours post inoculation. Relative infection (%) was calculated from infection of DMSO-treated cells. Each point in **(B)** and **(C)** represents the mean of triplicate samples from one experiment. Bars indicate the average of three independent experiments and error bars indicate SD.

TMPRSS2 expression allowed SARS-CoV-2-S protein-mediated entry to bypass the endosomal activation pathway. To test this possibility, TMPRSS2(-) and TMPRSS2(+) cells were treated with ammonium chloride prior to infection. As expected, the inhibitory effect of ammonium chloride was more robust on TMPRSS2(-) cells for both SARS1- and SARS2-PV (**Fig 2B**). Without TMPRSS2, more than 60% of the SARS2-PV entry was inhibited by ammonium chloride. For VSV-G pseudotypes, which are pH sensitive but do not utilize TMPRSS2 for proteolysis activation, the inhibition efficiency of ammonium chloride was moderate, and unaffected by TMPRSS2 expression. TMPRSS2-expressing cells treated by ammonium chloride has a 10-fold higher viral infection compared to the control group (TMPRSS2-negative cells with DMSO treatment), suggesting that for SARS-CoV-2 S-protein mediated entry, the TMPRSS-mediated activation pathway is much more efficient than the endosomal activation pathway. Moreover, in the presence of TMPRSS, ammonium chloride had only a modest inhibitory effect on SARS2-PV infection.

To determine the precise proteases involved in the SARS-CoV-2-S proteolysis activation, a panel of cysteine protease inhibitors was tested on TMPRSS2 (-) and TMPRSS2 (+) HEK293-T-ACE2 cells. Among these compounds, E64d and Z-III-FMK inhibit both cathepsin B and

cathepsin L; MDL281740 is a cathepsin-L inhibitor and CA-074 is a cathepsin-B inhibitor. In the absence of TMRPPS2 expression, E64d, Z-III-FMK, and MLD281740 showed robust inhibition of SARS2-PV (**Fig 2C**). These results suggest that, as observed with SARS1-PV, cathepsin L but not cathepsin B, facilitated SARS-CoV-2 infection. In contrast, when TMPRSS2 was expressed, the antiviral activity of these cathepsin inhibitors was largely abrogated. Collectively, these results suggest that in the absence of TMPRSS2 expression, endosomal cathepsin L is critical to viral entry mediated by SARS-CoV-2-S protein; however, when TMPRSS2 is expressed, the role of cathepsin L is markedly diminished.

## TMPRSS2 expression significantly attenuates the antiviral effect of hydroxychloroquine against SARS-CoV-2-S

Hydroxychloroquine can be used as a broad-spectrum antiviral drug against multiple viruses [34]. Although its exact antiviral mechanism remains unclear, it is well established that hydroxychloroquine accumulates within the acidic organelles such as endosomes [14]. As a weak base, it thereby increases the pH of endosomes, and subsequently interferes with the activities of pH-dependent endosomal protease. Because the antiviral effect of ammonium chloride on SARS-CoV and SARS-CoV-2 can be overcome by TMPRSS2 expression, we investigated whether TMPRSS2 expression also affects the antiviral activities of hydroxychloroquine. We observed that transient expression of TMPRSS2 in HEK293T-ACE2 cells resulted in higher TMPRSS levels than stable expression (**Fig 3A**), although only a low amount of plasmid was used for transient transfection (5 ng/well in 96-well plate). To demonstrate the effect of TMPRSS2 expression levels on the antiviral activities of hydroxychloroquine, both transient (high) and stable (low) TMPRSS2 HEK293T-ACE2 cells were used as targets. For HEK293-T-ACE2 cells transiently expressing the control plasmids (**Fig 3B**), hydroxychloroquine potently inhibited viral entry of SARS1- and SARS2-PV ($IC_{50}$ = 1.55 μM and 1.62 μM, respectively), and modestly reduced VSV-G-mediated cell entry. These $IC_{50}$ values are consistent with a previous cell culture study, using replicative SARS-CoV-2 on Vero cells, which express low level of TMPRSS2 [10]. However, when TMPRSS2 was expressed, hydroxychloroquine-mediated inhibition of SARS2-PV was substantially attenuated. The $IC_{50}$ of hydroxychloroquine against pseudotyped SARS-CoV-2 was increased by 5- to 60-fold for low and high TMPRSS2-expressing cells, respectively (**Fig 3B**). SARS1-PV also became more resistant to hydroxychloroquine with TMPRSS2 expression. Its $IC_{50}$ was increased by 20-fold with high TMPRSS2 expression, although no difference was observed with low TMPRSS2 expression. As a control, TMPRSS2 expression did not affect the inhibition of VSV-PV. These results indicate that TMPRSS2 contributes more strongly to SARS-CoV-2 infection than to SARS-CoV-1. To exclude the possibility of our observation being an artifact from overexpressing TMPRSS2, we tested the antiviral activity of hydroxychloroquine on human kidney (Vero) and lung epithelial cells (H1975, H1299, Calu-3), which express different level of TMPRSS2 (**Fig 3C**). We established H1975-ACE2 and H1299-ACE2 stable cell lines to acquire sufficient infection for quantitative analysis. Lentiviruses pseudotyped with SARS-CoV-1, SARS-CoV-2 S, and VSV G proteins were used for infection. Consistent with the results produced in HEK293T-ACE2 cells, the inhibitory efficiency of hydroxychloroquine negatively correlated with the expression level of TMPRSS2 for both SARS-CoV-1 and SARS-CoV-2, but not for VSV.

## Suppression of TMPRSS2 restores the antiviral efficiency of hydroxychloroquine

To confirm that the antiviral effect of hydroxychloroquine on SARS-CoV-1 and SARS-CoV-2 is masked by TMPRSS2 activities, we next explored whether suppression of TMPRSS2 could

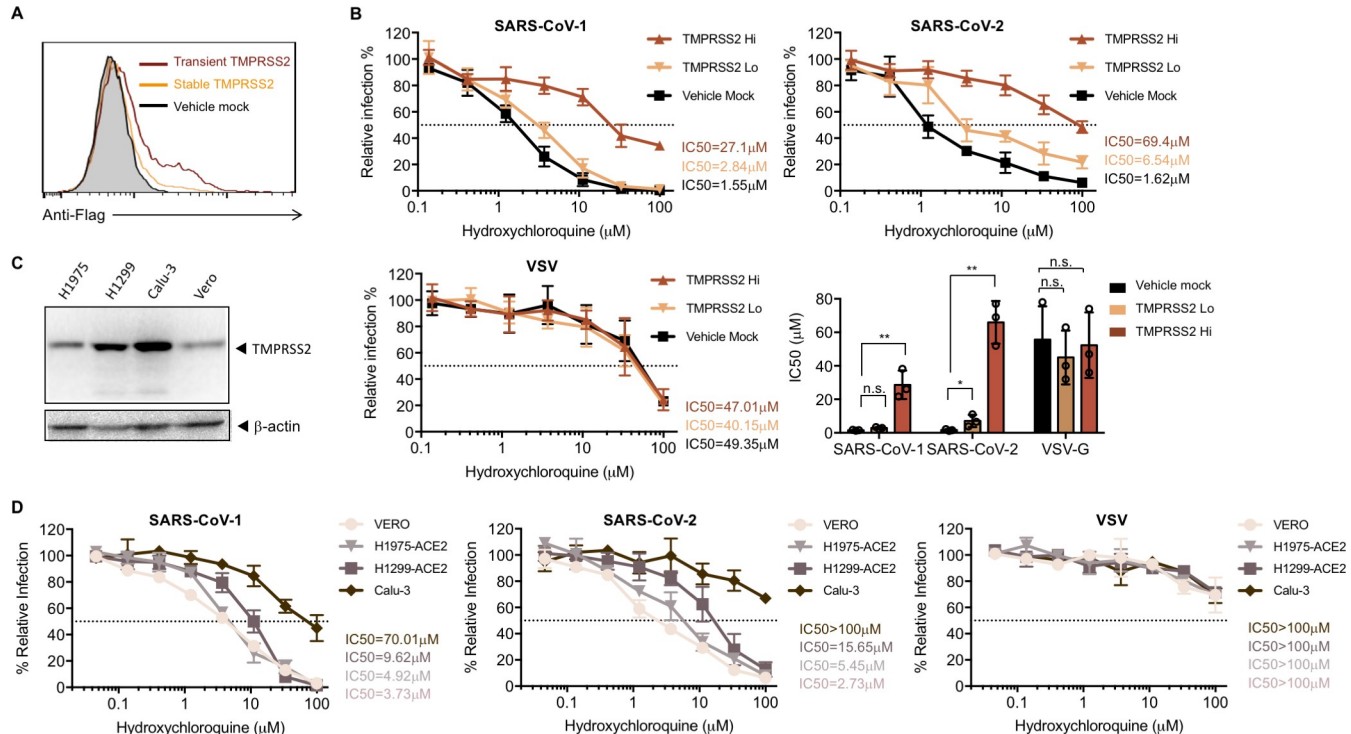

**Fig 3. Antiviral effect of hydroxychloroquine is dependent on TMPRSS2 expression. (A)** Cell-surface staining was performed by detecting the flag tag at the C-terminal of TMPRSS2 to validate the overexpression of TMPRSS2. A stable cell line of 293T-ACE2 cells was generated to express TMPRSS2 (orange line). 293T-ACE2 cells were transiently transfected with a vector control (black line), or TMPRSS2 (red line). The 293T/ACE2/TMPRSS2 stable cell line had much lower expression of TMPRSS2 compared to 293T-ACE2 transiently transfected with TMPRSS2 plasmids, and thus were referred as TMPRSS2 Lo and TMPRSS2 Hi, respectively. Shown is a representation of flow cytometry data from two independent experiments. **(B)** 293-ACE2 cells with different levels of TMPRSS2 were treated with hydroxychloroquine or DMSO before virus inoculation. The results are presented as a percentage of infection of DMSO-treated cells ($\approx$ 250,000 RLU for all three pseudotypes from TMPRSS-expressing cells.) Shown are representative plots of the mean value ($\pm$SD) of triplicate samples from the viral entry inhibition assay. The bar graph is a summary of $IC_{50}$ from three independent experiments. Each point represents the IC50 calculated from one experiment. Unpaired Student's t-test was used to assess the statistical significance of the difference between $IC_{50}$ on mock transfected cells (TMPRSS2-negative) and TMPRSS2-positive cells. (**: $P < 0.01$. *: $P < 0.05$. n.s.: $P > 0.05$.) **(C)** Shown is a representative western blot (of two independent experiments performed) reflecting expression level of TMPRSS2 in different cell lines. H1299, H1975 and Calu-3: human non-small cell lung cancer cells; Vero: kidney epithelial cells. β-Actin served as loading controls. **(D)** Lentiviruses pseudotyped with SARS-CoV-1, SARS-CoV-2 S, and VSV G proteins were used for infection. Stable cells lines of H1299 and H1975 over-expressing ACE2 were generated to allow efficient infection. Drug treatment, virus inoculation, and luciferase measurement were the same with the procedures in **(B)**. Shown are the mean value ($\pm$SD) of triplicate samples from three independent experiments.

rescue the antiviral efficiency of hydroxychloroquine. Camostat, a clinically proven drug that specifically inhibits TMPRSS2, was tested in combination with hydroxychloroquine on TMPRSS2-expressing cells inoculated with SARS1-, SARS2-, or VSV-PV (**Fig 4A**). We examined the effect of varying concentration of hydroxychloroquine in the presence of fixed (10 μM) amounts of camostat and a cysteine protease inhibitor E64d (inhibits cathepsin-L). Consistent with the presumption that E64d and hydroxychloroquine redundantly interferes with endosomal activation of the S proteins, E64d only modestly enhanced the inhibitory efficiency of hydroxychloroquine for SARS2-PV (changing the hydroxychloroquine $IC_{50}$ from 70.5 μM to 22.4 μM, **Fig 4B**) and SARS1-PV (19.6 μM to 9.63 μM). In contrast, the same fixed amount of camostat had a more than 20-fold impact on hydroxychloroquine inhibition of SARS2-PV (changing the $IC_{50}$ from 70.5 μM to 3.2 μM), but much less impact on SARS1-PV (19.6 μM to 10.57 μM). Notably, hydroxychloroquine alone more strongly inhibited SARS1-PV than did camostat, whereas the reverse was true for SARS-CoV-2, where camostat alone inhibited much more efficiently than hydroxychloroquine (**Fig 4B**). These data again

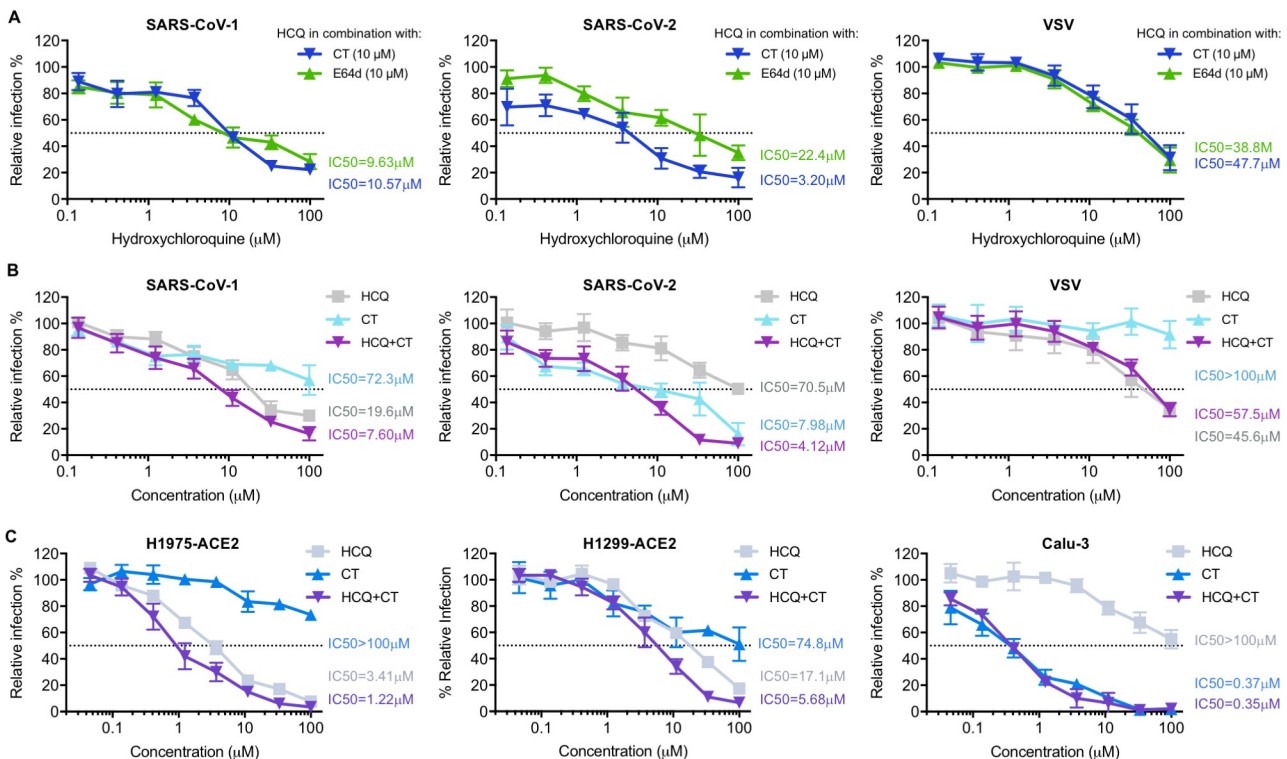

**Fig 4. Suppression of TMPRSS2 restores the antiviral efficiency of hydroxychloroquine. (A)** Hydroxychloroquine (HCQ) and camostat (CT) were tested on 293T-ACE2 cells transiently expressing TMPRSS2 prior to infection. HCQ was serially diluted with complete media containing 10 μM of camostat (CT, blue solid line), or 10 μM of E64d (cyan solid line), respectively. The antiviral efficiency of hydroxychloroquine in combination of each inhibitor was compared. **(B-C)** The antiviral efficiency of HCQ and CT alone, and their combination, were tested on **(B)** 293T-ACE2 cells and **(C)** other human lung cell lines (H1975-ACE2, H1299-ACE2, and Calu-3). Cells were challenged with retroviruses pseudotyped with SARS-CoV-2, SARS-CoV-1 S proteins, and VSV-G in **(B)**, and with lentivirus pseudotyped with SARS-CoV-2 S proteins in **(C)**, after drug or DMSO treatment. Luciferase activity was measured at 48 hours post inoculation. The average of three independent experiments conducted with triplicates is shown in **(A–C)**. Error bars indicate SD. Relative infection (%) was calculated from infection of DMSO-treated cells.

demonstrate that SARS2-PV are more responsive to TMPRSS2 inhibition than are SARS1-PV, and that TMPRSS2 activity must be suppressed for hydroxychloroquine to be efficient, and thus the combination of both drugs inhibited SARS2-PV more effectively than either drug alone. Similar results were reproduced in multiple human lung epithelial cell lines including H1975-ACE2, H1299-ACE2, and Calu-3 cells (**Fig 4C**), suggesting that this drug combination could also be effective in epithelial cells *in vivo*.

## An S-protein furin-cleavage site increases reliance on TMPRSS2 expression

The data from experiments above consistently indicated that SARS-CoV-2 is more dependent on TMPRSS2 than SARS-CoV-1. Because a major difference of SARS-CoV-2 S protein from that of SARS-CoV-1 is the presence of a furin-cleavage site at its S1/S2 boundary, we investigated whether the reliance on TMPRSS2 expression can be changed by knocking out the furin-site (FKO) in SARS-CoV-2 S protein and by inserting a furin cleavage site in SARS-CoV-1 S protein at the S1/S2 boundary. In addition, previous studies suggest that the furin site renders SARS-CoV-2 S protein relatively unstable, a phenotype partially restored by a naturally occurring D614G mutation in the S1 domain [35]. We therefore investigated whether this D614 mutation would render it less dependent on TMPRSS2. The infectivity of pseudoviruses bearing the FKO and D614G S-protein variants was compared to wildtype SARS-CoV-2 S

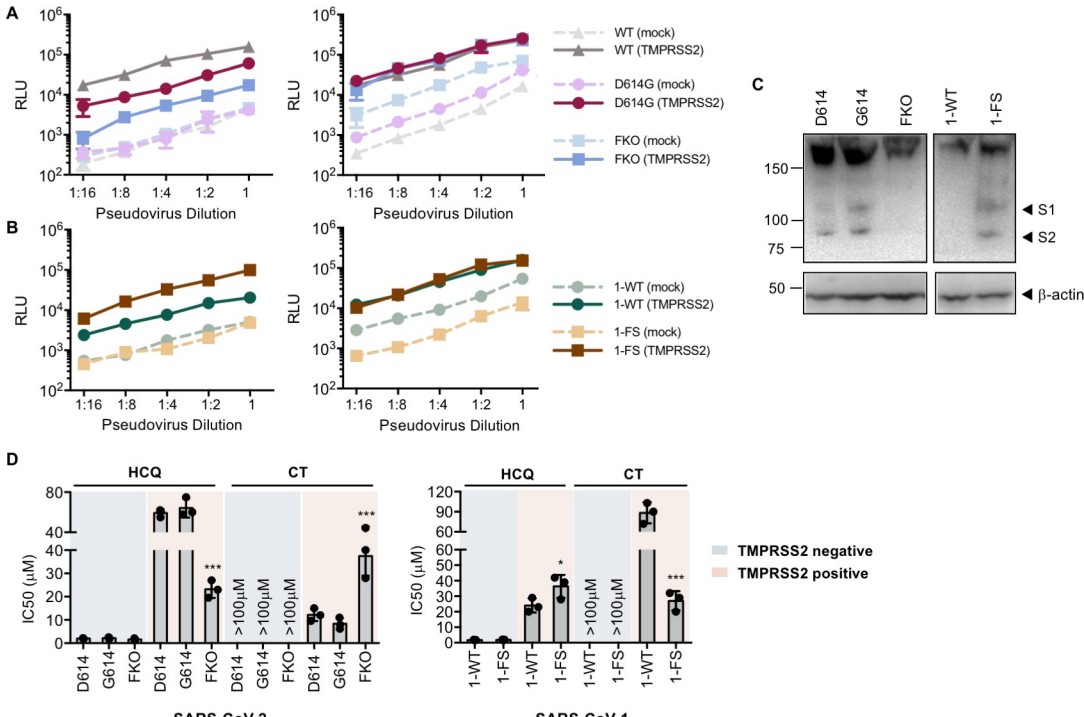

**Fig 5. Furin cleavage in the virion producer cell correlates with TMPRSS2 dependence. (A-B)** The infectivity of SARS-CoV-1 and SARS-CoV-2 and their mutants on 293-ACE2 cells was compared with or without overexpression of TMPRSS2. The infection assay was performed as described in **Fig 2A**. Retroviruses were pseudotyped with S proteins of SARS-CoV-2 wildtype (WT), D614G S-protein variant, SARS-CoV-2 with furin-site knockout (FKO), and (B) SARS-CoV-1 WT (1-WT), and SARS-CoV-1 with the furin site derived from SARS-CoV-2 (1-FS). Pseudovirus titers were adjusted so that they were equivalent in the absence (left panel) or presence (right panel) of TMPRSS2. Shown is a representative plot of the mean (±SD) of duplicate samples from at least two independent experiments. **(C)** Western blot analysis of S protein cleavage of SARS-CoV-2-S WT (D614) and variants (G614 and FKO), and SARS-CoV-1 WT and its variant with furin site addition. Antibody detected the flag tags at the N- and C-termini of S proteins. β-Actin served as loading controls. Black arrow heads indicate bands corresponding to the S1 and S2 subunits from cleavage of S proteins. Shown are representative blots from two experiments. **(D)** The effect of TMPRSS2 on the antiviral efficiency of hydroxychloroquine (HCQ) and camostat (CT) was compared. Retrovirus pseudotyped against S proteins described in **(C)** after drug or DMSO treatment. Luciferase activity was measured at 48 hours post inoculation. Relative infection (%) was calculated from infection of DMSO-treated cells. Statistical significance between wildtypes and mutants was tested by two-way ANOVA with Dunnett's posttest. (***: $P < 0.001$. *: $P < 0.05$).

protein on TMPRSS2-positive and TMPRSS2-negative cells. As shown in **Fig 5A**, when pseudovirus titers were adjusted to be equivalent in the absence of TMPRSS2 (left panel), entry mediated by the wildtype S protein was most enhanced by TMPRSS2 expression (up to 100-fold), and that mediated by the FKO was least affected (10-fold). The D614G variant showed an intermediate phenotype (30-fold), consistent with its greater stability than wildtype S protein. To confirm this results, we then adjusted pseudovirus titers to be equivalent in the presence of TMPRSS2 (right panel). Similar comparison experiments were performed for wildtype SARS-CoV-1 and the furin inserted mutant (**Fig 5B**). Entry mediated by the furin-inserted S protein was most enhanced by TMPRSS2 expression (up to 40-fold) than that mediated by the wildtype (up to 10-fold). To verify that the introduced mutations indeed altered proteolysis at the S1/S2 boundary, immunoblot analysis was conducted on HEK293T cells expressing S proteins containing Flag tags at both their N- and C-termini (**Fig 5C**). Consistent with our previous studies [35], more release or shedding of S1 domain (reflected as a weaker signal from S1 band) was observed from the wildtype S protein than from the D614G variant, suggesting that the D614G variant was more stable. By introducing the furin site, the S protein

of SARS-CoV-1 was cleaved in a similar way with that of the SARS-CoV-2 D614G variant. No proteolytic cleavage was observed in the S proteins of SARS-CoV-2 FKO or SARS-CoV-1 wild-type, confirming that the furin-recognition site at the S1/S2 junction was necessary for cleaving the S proteins in virus-producing cells. The drug sensitivity to hydroxychloroquine and camo-stat of pseudoviruses harboring these S-protein mutants was also compared to those harboring the wildtype S proteins on both TMPRSS2-negative and TMPRSS2-positive cells (**Fig 5D**). In the absence of TMPRSS2, entry mediated by all S-protein variants was similarly sensitive to hydroxychloroquine, and, as expected, unaffected by camostat. However, in the presence of TMPRSS2, the FKO S protein of SARS-CoV-2, compared to its wildtype, was more inhibited by hydroxychloroquine and less inhibited by camostat. And the furin-inserted S protein of SARS-CoV-1, compared to its wildtype, became less sensitive to hydroxychloroquine and more to camostat. Collectively, the furin-cleavage site of S protein determines reliance on TMPRSS2, sensitivity to camostat, and resistance to hydroxychloroquine.

## Discussion

It has been previously demonstrated that SARS-CoV-1 and SARS-CoV-2 entry into cells depends on the expression on two somewhat redundant proteases, namely cathepsin L located in acidic cellular compartments, and TMPRSS2, expressed on the plasma membrane [24,36]. It is further understood from these and other observations that, unlike pH-dependent viruses such as influenza A virus, conformational changes of the spike protein of coronaviruses are not dependent on pH [15]. Rather, proteolytic activation by cathepsin L is dependent on endo-somal acidification, and thus elevating endosomal pH prevents cathepsin-L-mediated entry. However, the TMPRSS2-mediated entry pathway, in which S-protein is presumed to be acti-vated at the plasma membrane, is not affected by pH (**Fig 1**). Here we extend these observa-tions in two directions. First we show that SARS-CoV-2 is more dependent on TMPRSS2 than is SARS-CoV-1, and that this difference can largely be explained by the presence of a furin cleavage site in the SARS-CoV-2 S protein. Second, we show that TMPRSS2 expression over-comes the antiviral effect of hydroxychloroquine, thus providing a mechanistic explanation for its poor therapeutic efficacy against SARS-CoV-2 despite encouraging cell-culture results.

We present multiple lines of evidence that SARS-CoV-2 is more sensitive to the presence of TMPRSS2 than is SARS-CoV-1. With PV titers adjusted so that SARS1-PV and SARS2-PV infection were comparable in the presence of TMPRSS2, SARS2-PV transduced cells markedly less efficiently in its absence (**Fig 2A**). In the presence of TMPRSS2, SARS1-PV are more sensi-tive than SARS2-PV to inhibitors of endosomal acidification such as ammonium chloride (**Fig 2B**) and hydroxychloroquine (**Fig 3B and 3C**). Specifically, the $IC_{50}$ of hydroxychloroquine for SARS1-PV was three-fold lower than for SARS2-PV with high TMPRSS2 expression, while their sensitivities to hydroxychloroquine are equivalent in the absence of TMPRSS2 (**Figs 3B and 4B**). By contrast, SARS2-PV were 12-fold more sensitive to the TMPRSS2 inhibitor camo-stat (**Fig 4B**). Although TMPRSS2-mediated pathway is preferred over the cathepsin-L-medi-ated pathway for both SARS-CoV-1 and -2, these data indicate that SARS-CoV-1 utilizes the cathepsin-L-pathway more efficiently than SARS-CoV-2, whereas SARS-CoV-2 is more dependent on TMPRSS2 than SARS-CoV-1.

What accounts for this difference? The most obvious difference between the SARS-CoV-1 and -2 S proteins is the presence of a polybasic or furin-cleavage site at the boundary between the S1 and S2 domains [21]. This furin site is present in other human coronaviruses, for exam-ple MERS-CoV and HCoV-OC43 [37,38], but it has not previously been observed in any of the many SARS-like coronaviruses identified in bats [39]. Thus, the S1/S2 boundary of the SARS-CoV-2 S protein, but not that of SARS-CoV-1, is cleaved at this site in the virus-

producing cells. Indeed, when the SARS-CoV-2 furin-cleavage site is ablated, the mutant PV is less impacted by the addition or removal of TMPRSS2 compared to the wildtype (**Fig 5A**), and these pseudoviruses are relatively more sensitive to hydroxychloroquine-mediated inhibition, and less sensitive to camostat (**Fig 5D**).

Thus furin-cleavage in the virus-producing cell correlates with greater dependence on TMPRSS2, and lower dependence on cathepsin L. If one assumes that several proximal S-protein trimers must be fully cleaved and activated to mediate fusion, this difference is relatively easy to understand. Specifically, furin cleavage in virus-producing cells reduces the number of required proteolytic events in the target cell, and thus in many cases TMPRSS2 alone can fully activate fusion. In the case of SARS-CoV-1, or when the furin-cleavage site is replaced by one cleaved only by TMPRSS2, more target-cell cleavage events are required. In such cases, to complete the proteolytic activation of sufficient numbers of S proteins, cathepsin L-mediated proteolysis, and therefore viral endocytosis and endosomal acidification, are necessary.

However our data also make clear that, in the absence of TMPRSS2, SARS-CoV-2 utilizes cathepsin L less efficiently than SARS-CoV-1. For example, when titers are adjusted so that infections by SARS1- and SARS2-PV are identical in the presence of TMPRSS2 (**Fig 1A**), their efficiencies are markedly different in its absence. This difference can perhaps be explained by the relative instability of the wildtype SARS-CoV-2 S protein [16,40] in two ways. First, the furin-cleaved SARS-CoV S protein has been shown to prematurely shed [35], resulting in fewer S proteins per virion and pseudovirion. The resulting lower density of S proteins per virion may impair the ability of cross-linked ACE2 to promote endocytosis of the virus. Alternatively, the less stable S protein may be further destabilized in the acidifying endosome, disrupting the ordered steps of fusion mediated by proteolytic activation and conformational transitions of S2. Thus the more stable D614G SARS-CoV-2 S protein variant [35] is modestly less affected by the presence and absence of TMPRSS2 than the wildtype (D614) S protein (**Fig 5A**).

The greater dependence of SARS-CoV-2 on TMPRSS2 has an immediate implication for the treatment of COVID-19. Specifically, it implies that inhibitors of endosomal acidification will have less impact on SARS-CoV-2 in the presence of TMPRSS2. We show here that indeed, TMPRSS2 helps bypass the hydroxychloroquine-mediated inhibition of SARS2-PV infection. Most physiologically relevant target cells in the body, include type II pneumocytes and ciliated nasal epithelial cells express TMPRSS2. Thus the potent inhibition of SARS-CoV-2 by hydroxychloroquine in Vero E6 cells, where TMPRSS2 is largely absent, overestimated its potency by 10- to 40-fold, depending on TMPRSS2 expression (**Fig 3A and 3C**). However, our results suggest that some efficacy of hydroxychloroquine could be restored if TMPRSS2 is inhibited by camostat (**Fig 4A and 4B**), an observation directly relevant to clinical trials, for example NCT04338906, combining the two inhibitors.

In summary, we show that the inhibitory effect of hydroxychloroquine on SARS-CoV-2 entry is attenuated by TMPRSS2, thus explaining its limited clinical efficacy. We further show that SARS-CoV-2 is more dependent on TMPRSS2 and less dependent on cathepsin L than SARS-CoV-1, and that these differences are likely due to the presence of a furin-cleavage site in the S protein of SARS-CoV-2. Finally, we show inhibition of both TMPRSS2 and cathepsin L may be necessary to fully block virus entry in cells that express both proteases.

## Materials and methods

### Plasmid

Plasmids encoding TMPRSS2 and the control empty vectors were purchased from OriGene (PS100001). DNA sequence of SARS-CoV-2 S protein (GenBank YP_009724390) was codon-

optimized and synthesized by Integrated DNA Technologies (IDT), and was cloned subsequently into pCAGGS vector using In-Fusion HD Cloning Kit (Takara Bio USA) according to manufacturer's instructions. FKO S protein of SARS-CoV-2 was created by replacing -RRAR- at S1/S2 junction with -SRAS-; furin-inserted S protein of SARS-CoV-1 was created by inserting -PRRA- to the S1/S2 boundary. SARS-CoV-2 D614G and furin-site mutated S proteins were made by site-directed mutagenesis.

## Pseudovirus production

Murine Leukemia Viruses (MLV) pseudotyped with variant spike or envelope proteins were generated as described before [41]. Briefly, HEK293T cells were co-transfected with three plasmids, pMLV-gag-pol, pQC-Fluc and pCAGGS-SARS2-S-cflag or pcDNA3.1-SARS1-S or pCAGGS-VSV-G, and the medium was refreshed after 6h incubation of transfection mix. The supernatant with produced virus was harvested 72h post transfection and clarified by passing through 0.45μm filter. Clarified viral stocks were supplemented with HEPES with the final concentration of 10mM and stored at -80˚C for long-term storage. Lentiviral pseudoviruses were produced in the same way by co-transfection of the spike/envelop plasmids and the HIV-1 NL4-3 ΔEnv luciferase reporter vector (1:5 ratio). The reporter vector was obtained from Dr. Nathaniel Landau through the NIH AIDS Reagent Program. The SARS-CoV-2 spike protein plasmid was a gift from Raffaele De Francesco (Addgene plasmid #155297).

## Cell culture and stable cell line

The HEK293T, H1299 and H1975 cell lines expressing human ACE2 (ACE2) were created by transduction with produced VSV G protein-pseudotyped MLV containing pQCXIP-myc-ACE2-c9 as described in the section above. The parental cells were transduced with generated MLV virus, and the ACE2 cell lines were selected and maintained with medium containing puromycin (Sigma-Aldrich). ACE2 expression was confirmed by immunofluorescence staining using mouse monoclonal antibody against c-Myc antibody 9E10 (Thermo Fisher) and goat-anti-mouse FITC (Jackson ImmunoResearch Laboratories).

HEK293T/ACE2/TMPRSS2 stable cell line was also constructed by transducing 293T-ACE2 cell line with MLV pseudovirus made by cotransfection of pMLV-gag-pol, pQCXIB-TMPRSS2-Flag and pCAGGS-VSV-G at 3:2:1 ratio into 293T cells. 293T, Vero, and Calu-3 cells were maintained in DMEM (Life Technologies), and H1299 and H1975 cells in RPMI 1640 (Gibco), at 37˚C in a 5% $CO_2$-humidified incubator. Growth medium were supplemented with 2 mM Glutamax-I (Gibco), 100 μM non-essential amino acids (Gibco), 100 U/mL penicillin and 100 μg/mL streptomycin (Gibco), and 10% FBS (Gibco). For all the ACE2 stable cell lines, 1 μg/mL of puromycin was added to the growth medium to maintain expression of ACE2. For 293T/ACE2/TMPRSS2, 1 μg/mL of puromycin and 10 μg/mL blasticidin was added to the growth medium.

## Pseudovirus infection

HEK293T-ACE2 cells were seeded at 30% density in poly-lysine (Sigma-Aldrich) pre-coated 96-well plates 12–15 hours prior to transfection. Cells in each well were then transfected with 0.3 μL of lipofectamine 2000 (Life Technologies) in complex with 5 ng of a vector control plasmid or a plasmid encoding TMPRSS2. Cell culture medium was refreshed at 6 hours post transfection. Additional 18 hours later, cells were infected with pseudovirus diluted in 100 μL of culture medium containing 2% FBS (Gibco). Cells were spin-infected at 4˚C for 30 min at 3000xg to allow virus-binding to cells, followed by 2 hours of incubation at 37˚C. After incubation with virus, supernatant was removed, and each well was replenished with 200 μL of fresh

media containing 2% FBS. Same infection procedures were applied on other cell lines without the plate coating and transfection steps.

## Cell surface expression and S protein analysis

To measure surface TMPRSS2 expression of 293T-ACE2 transiently transfected with TMPRSS2 and the stable cell line 293T/ACE2/TMPRSS2, cells were detached by 1mM EDTA in PBS and then stained by 2 ug/ml of anti-Flag M2 antibody (Sigma-Aldrich, F1804) and 2 μg/ml of goat anti-mouse IgG (H+L) conjugated with Alexa 647 (Jackson ImmunoResearch Laboratories, #115-606-146). Flow cytometry analysis was done using Accuri C6 (BD Biosciences). To measure the endogenous TMPRSS2 expression of Vero, H1299, H1975 and Calu-3 cells, cells were permeabilized with PBS including 0.5% Triton X-100 (Sigma-Aldrich) at room temperature for 10 min, and detected by 2 μg/ml monoclonal rabbit Anti-TMPRSS2 antibody [EPR3861] (Abcam, ab92323) and goat anti-rabbit IgG conjugated with HRP (Sigma-Aldrich, A0545).

To determine the cleavage of S proteins, 293T cells were transfected with 2 μL of lipofectamine 2000 (Life Technologies) in complex with 1 μg plasmid expressing the indicated S protein variant. Cells were harvested for western blot analysis 48 hours post transfection. Cells were permeabilized with PBS including 0.5% Triton X-100 (Sigma-Aldrich) at room temperature for 10 min, and detected by 1 μg/ml anti-Flag M2 antibody (Sigma-Aldrich, F1804) and goat anti-mouse IgG (Fab only) conjugated with HRP (Sigma-Aldrich, A9917).

## Luciferase assay for viral entry and inhibition of viral entry

At 48 hours post infection, cells were lysed in wells and subjected to *Firefly* luciferase assays. Viral entry was determined using Britelite Plus (PerkinElmer), and luciferase expression was measured using a Victor X3 plate reader (PerkinElmer). For experiments testing drug-mediated inhibition, target cells were treated by the respective chemicals or DMSO diluted in 100 μL of media containing 2% FBS to the final indicated concentrations. All compounds we tested were purchased from Sigma-Aldrich: ammonium chloride (A9434), E64-d (E8640), Z-III-FMK (C8984); MDL281740 (M6690), CA-074 (C5732), hydroxychloroquine (PHR1782), camostat (SML0057). After incubation for 2 hours at 37˚C, supernatant was removed prior to virus transduction.

## Statistical analysis

Data expressed as mean values ± S.D., and all statistical analysis was performed in GraphPad Prism 7.0 software. $IC_{50}$ of drugs was analyzed using default settings for log(inhibitor) vs. normalized response method. Statistical difference was determined using non-paired Student's t-test or two-way ANOVA with Dunnett's posttest. Differences were considered significant at $P < 0.05$.

## Supporting information

**S1 Data. Excel spreadsheets containing the underlying numerical data to generate Figs 2A, 2B, 2C**, **3B, 3D**, **4A, 4B, 4C**, **5A, 5B, and 5D**.
(XLSX)

## Author Contributions

**Conceptualization:** Tianling Ou, Michael Farzan.

**Funding acquisition:** Michael Farzan.

**Investigation:** Tianling Ou, Huihui Mou, Lizhou Zhang, Amrita Ojha.

**Methodology:** Tianling Ou, Huihui Mou, Lizhou Zhang.

**Supervision:** Hyeryun Choe, Michael Farzan.

**Visualization:** Tianling Ou, Huihui Mou.

**Writing – original draft:** Tianling Ou, Huihui Mou, Michael Farzan.

**Writing – review & editing:** Tianling Ou, Huihui Mou, Lizhou Zhang, Amrita Ojha, Hyeryun Choe, Michael Farzan.

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
