## [Decision Letter · Decision Letter 0]

21 Aug 2020

Dear Dr. Farzan,

Thank you very much for submitting your manuscript "Hydroxychloroquine-mediated inhibition of SARS-CoV-2 entry is attenuated by TMPRSS2" for consideration at PLOS Pathogens. As with all papers reviewed by the journal, your manuscript was reviewed by members of the editorial board and by several independent reviewers. In light of the reviews (below this email), we would like to invite the resubmission of a revised version that addresses the common overriding concerns outlined in the reviewers' comments.

I had invited 3 experts who agreed to review this manuscript. This pandemic has made it difficult to secure the last review in a timnely fshion, so only two reviews are attached.  Both reviewers recognized the significance of the topic, but also that the rigor was lacking to some extent. This must be addressed (number of repeats, statistics, etc.). Please try to meet their major concerns more than half-way. I would add that since the study hinges quite a bit on proteolytic activation of SARS-CoV-2 entry, biochemical support  for how HCQ modulates TMPRSS2-dependent vs independent cleavage and entry should be provided. This would greatly enhance the imnpact of your paper, as the field is now mature enough to link proteolytic activation with biochemical evidence of cognate cleavage.  

We cannot make any decision about publication until we have seen the revised manuscript and your response to the reviewers' comments. Depending on how thorough you address the reviewer's comments, your revised manuscript may be sent to reviewers for further evaluation. We try to avoid multiple rounds of review, so I urge you to provide a well-copnsidered revision that will enablke me to render a decisiopn without the further delay of re-review. 

Sincerely,

Benhur Lee

Section Editor

PLOS Pathogens

Benhur Lee

Section Editor

PLOS Pathogens

Kasturi Haldar

Editor-in-Chief

PLOS Pathogens

orcid.org/0000-0001-5065-158X

Michael Malim

Editor-in-Chief

PLOS Pathogens

orcid.org/0000-0002-7699-2064

Reviewer's Responses to Questions

**Part I - Summary**

Reviewer #1: In the manuscript by Ou et al, the authors assess the ability of hydroxychloroquine to inhibit entry of SARS-CoV-2. They demonstrate that hydroxychloroquine has limited activity against TMPRSS2-mediated entry but is effective against Cathepsin L-mediated entry. They further demonstrate that hydroxychloroquine is synergistic with the TMPRSS2 inhibitor Camostat. Unraveling the mechanism and potential utility (or lack thereof) of entry inhibitors such as hydroxychloroquine is of significant interest for the treatment of COVID-19. However, enthusiasm for the manuscript is tempered by the complete reliance on pseudovirus assays and transgenic cell lines and questionable experimental rigor (as it was poorly described)

Reviewer #2: The S proteins of SARS-CoV-1 and -CoV-2 are dependent on proteolytic cleave, either by TMPRSS2 at the plasma membrane or by cathepsin L in low pH endosomes, before they can mediate fusion with the target cell membrane and viral entry. Using cathepsin inhibitors, cell lines differing in TMPRSS2 expression, the TMPRSS2 inhibitor camostat, and varying concentrations of hydroxychloroquine, Ou et al. show that SARS-CoV-2 is more dependent on TMPRSS2 for S protein cleavage and entry than SARS-CoV-1. As a consequence, SARS-CoV-2 is more resistant than SARS-CoV-1 to the effects of hydroxychloroquine, which inhibits endosomal acidification and low pH cathepsin-mediated activation of the S protein. The dependence of SARS-CoV-2 on TMPRSS2 also correlates with the presence of a polybasic cleavage site in S for furin-mediated cleavage in the producer cell.

The data showing that SARS-CoV-2 is more dependent on TMPRSS2 than SARS-CoV-1, and is therefore less sensitive to hydrochoroquine, is convincing and has important clinical implications, since it helps to explain why hydrochoroquine was able to inhibit SARS2 infectivity in Vero E6 lacking TMPRSS2, but ultimately failed to confer a beneficial effect in clinical trials. However, the relationship between furin-mediated cleavage of the SARS2 S protein and greater dependence of TMPRSS2 is less complete, since it only compares the infectivity of wild-type, furin-site knockout (FKO) and D614G viruses. Ideally, one would like to see biochemical confirmation of the effects of these mutations on S protein sensitivity to furin and/or an increase in TMPRSS2 utilization by the SARS1 S protein after the introduction of a polybasic furin cleavage site.

**Part II – Major Issues: Key Experiments Required for Acceptance**

Reviewer #1: Major:

1) The entire manuscript depends on pseudovirus. While a powerful system, it may not reflect the stoichiometry, conformation, or ratio of cleaved:uncleaved spike on native SARS-CoV-2 virions. Validation of key findings herein with native SARS-CoV-2 virions (WT, D614G, and furin mutant) is critical. The WT and D614G viruses are readily available and the experiments are straightforward with the reagents in hand.

2) The entire manuscript exclusively utilizes 293T cells overexpressing key entry molecules ACE2 and TMPRSS2. Key experiments should be reproduced in a physiologically relevant system such as a primary human airway culture system (or ACE2 transgenic mice).

3) Each figure legend should state how many independent experiments were performed and with how many replicates per experiment. It appears that some figures represent only a single experiment with multiple technical replicates. If true, this would be unacceptable for publication.

Reviewer #2: 1. Differences in high versus low TMPRSS2 expression in stable 293T-ACE2 cell lines (Fig. 3) need to be supported by flow cytometry and/or western blot data comparing differences in expression.

2. The effects of mutations in the polybasic cleavage on sensitive to furin (Fig. 5) should be supported by western blots.

3. Does the introduction of the furin cleavage site into the SARS-CoV-1 S protein increase its ability to use TMPRSS2?

**Part III – Minor Issues: Editorial and Data Presentation Modifications**

Reviewer #1: Minor

4) Fig 2A, the authors should show a mock virus to reveal the limit of detection of the assay.

5) The figure 2 title says “inhibiting endosomal proteases does not reduce SARS2 infectivity in TMPRSS2 cells”. The E64D and ZIIIFMK data refute this as all 10uM data points are below the 100% line. This reviewer appreciates that the magnitude of reduction is very modest (and arguably not biologically meaningful) but the figure title as written not accurate given the data.

6) Histograms should show individual data points and not simply box and whiskers.

7)Last line of intro should be changed to “may reflect” as the failure of hydroxychloroquine in vivo is unclear.

8)Needs careful proofreading as there are numerous typos

Reviewer #2: 1. The manuscript would benefit from careful editing to correct numerous instances of misspelled or missing words (e.g. “facitiatea”; “hydroxychloroquine alone inhibited SARS1-PV more than camostat”).

2. Statements in the Introduction referring to hydroxychloroquine clinical trials and the EC50 of hydroxychloroquine in Vero E6 are missing references. This is also true of the statement that D614G partially restores S protein stability in the Results.

3. In the legend for Figure 5, the first line should refer to Fig. 2A (not Fig. 1A).

PLOS authors have the option to publish the peer review history of their article (what does this mean?). If published, this will include your full peer review and any attached files.

Reviewer #1: No

Reviewer #2: No
---

## [Editor Report · Decision Letter 1]

12 Nov 2020

Dear Dr. Farzan,

Thank you very much for submitting your manuscript "Hydroxychloroquine-mediated inhibition of SARS-CoV-2 entry is attenuated by TMPRSS2" for consideration at PLOS Pathogens. As with all papers reviewed by the journal, your revised manuscript was reviewed by members of the editorial board in light of the reviewers comments. 

Based on your good faith effort in responding to the reviewers' major concerns, and the additional data that extends the data of Hoffman et al, we are likely to accept this manuscript for publication, providing that you modify the manuscript according to the minor recommendations below:

In this age of rigor and reproducibility, please ensure that your materials and methods section is complete and comprehensive. For example, no information is given regarding where the major inhibitors central to the paper  were sourced (Camostat, E64d, HCQ, etc.) and the antibody used to detect endogenous TMPRSS2 in Fig. 3c appears to be missing. These are only the more obvious examples.      

Sincerely,

Benhur Lee

Section Editor

PLOS Pathogens

Benhur Lee

Section Editor

PLOS Pathogens

Kasturi Haldar

Editor-in-Chief

PLOS Pathogens

orcid.org/0000-0001-5065-158X

Michael Malim

Editor-in-Chief

PLOS Pathogens

orcid.org/0000-0002-7699-2064
---

## [Editor Report · Decision Letter 2]

3 Dec 2020

Dear Dr. Farzan,

We are pleased to inform you that your manuscript 'Hydroxychloroquine-mediated inhibition of SARS-CoV-2 entry is attenuated by TMPRSS2' has been provisionally accepted for publication in PLOS Pathogens.

Best regards,

Benhur Lee

Section Editor

PLOS Pathogens

Benhur Lee

Section Editor

PLOS Pathogens

Kasturi Haldar

Editor-in-Chief

PLOS Pathogens

orcid.org/0000-0001-5065-158X

Michael Malim

Editor-in-Chief

PLOS Pathogens

orcid.org/0000-0002-7699-2064
---

## [Editor Report · Acceptance letter]

12 Jan 2021

Dear Dr. Farzan,

We are delighted to inform you that your manuscript, "Hydroxychloroquine-mediated inhibition of SARS-CoV-2 entry is attenuated by TMPRSS2," has been formally accepted for publication in PLOS Pathogens.

Best regards,

Kasturi Haldar

Editor-in-Chief

PLOS Pathogens

orcid.org/0000-0001-5065-158X

Michael Malim

Editor-in-Chief

PLOS Pathogens

orcid.org/0000-0002-7699-2064